# Comparative Analysis of Rhizosphere and Endosphere Fungal Communities in Healthy and Diseased Faba Bean Plants

**DOI:** 10.3390/jof10010084

**Published:** 2024-01-22

**Authors:** Juan Li, Lu Hou, Gui Zhang, Liang Cheng, Yujiao Liu

**Affiliations:** 1Qinghai Academy of Agriculture and Forestry Sciences, Qinghai University, Xining 810016, China; 18293130625@163.com (J.L.); zhg-1195@163.com (G.Z.); liangcheng1979@163.com (L.C.); 2Key Laboratory of Agricultural Integrated Pest Management, Xining 810016, China; 3Key Laboratory of Qinghai Tibetan Plateau Biotechnology, Ministry of Education, Qinghai University, Xining 810016, China; 4State Key Laboratory of Plateau Ecology and Agriculture, Qinghai University, Xining 810016, China

**Keywords:** faba bean, fungal community, healthy and diseased, rhizosphere and endosphere, diversity, Illumina MiSeq

## Abstract

This study used the ITS approach based on Illumina MiSeq sequencing to assess the endosphere and rhizosphere fungal communities in healthy and diseased faba bean plants. The findings indicate that the most predominant phyla in all samples were Ascomycota (49.89–99.56%) and Basidiomycota (0.33–25.78%). In healthy endosphere samples, Glomeromycota (0.08–1.17%) was the only predominant phylum. In diseased endosphere samples, Olpidiomycota (0.04–1.75%) was the only predominant phylum. At the genus level, *Penicillium* (0.47–35.21%) was more abundant in rhizosphere soil, while *Paraphoma* (3.48–91.16%) was predominant in the endosphere roots of faba bean plants. Significant differences were observed in the alpha diversity of rhizosphere samples from different germplasm resources (*p* < 0.05). The fungal community structures were clearly distinguished between rhizosphere and endosphere samples and between healthy and diseased endosphere samples (*p* < 0.05). *Saccharomyces* was significantly enriched in diseased endosphere samples, whereas *Apiotrichum* was enriched in healthy endosphere samples. *Vishniacozyma* and *Phialophora* were enriched in diseased rhizosphere samples, while *Pseudogymnoascus* was enriched in healthy rhizosphere samples. Diseased samples displayed more strongly correlated genera than healthy samples. Saprotrophs accounted for a larger proportion of the fungal microbes in rhizosphere soil than in endosphere roots. This study provides a better understanding of the composition and diversity of fungal communities in the rhizosphere and endosphere of faba bean plants as well as a theoretical guidance for future research on the prevention or control of faba bean root rot disease.

## 1. Introduction

Faba bean (*Vicia faba* L.) is an herbaceous plant that belongs to the family Leguminosae and the genus *Vicia*. Because of its high nutritional value, faba bean has become a major food crop around the world [1]. However, the stress of biological and abiotic factors, especially diseases, has a significant effect on the yield and quality of faba bean. The main diseases of faba bean include chocolate spot, rust, ascochyta blight, alternaria leaf blight, cercospora leaf spot, downy mildew, and root rot [2,3,4,5].

Root rot disease is considered to be one of the factors limiting the cultivation and yield of faba bean [6]. Studies have shown that various fungi, such as *Fusarium* spp., *Rhizoctonia* spp., *Pythium* spp., *Phoma* spp., and *Aphanomyces* spp., can cause root rot in faba bean [7,8,9]. It has been reported that *Macrophomina phaseolina* and *Rhizoctonia solani*, which are soil-borne diseases, have a destructive effect on the growth of plants, including faba bean [4,10].

The relationship between microorganisms and host plants is very complex [11]. Root exudates attract soil microorganisms into the rhizosphere for nutrient transformation [12], and plant endophytic microorganisms can attract or filter microbes living in the rhizosphere [13]. Rhizosphere and endosphere microbes can promote plant growth, improve adaptability to the external environment, and resist various diseases and insects [14,15]. Some plant rhizosphere and endophytic communities have been studied, such as cotton [16], kiwifruit [17], and *Leymus chinensis* [18]. Changes in microflora can have an influence on plant health. However, the relationship between microbial communities in rhizosphere and endosphere of faba bean is still unclear.

Current studies have mostly focused on the screening and application of growth-promoting bacteria in rhizosphere of faba bean [4,19] and the variations in microbial communities after different treatments, such as storage time [20] and fermentation [21,22]. In addition, the microbial communities of intercropped or mixed crops, such as maize and faba bean and wheat and faba bean, have been studied [23,24]. These results indicated that intercropping is an effective method to reduce the incidence of root rot diseases in faba bean and increase the diversity of the soil microbial community [25]. Some biological agents have shown the potential to inhibit the root diseases of faba bean, including *Paenibacillus* spp., *Bacillus* spp., and *Trichoderma* spp. [26].

The core microbe and keystone microorganisms play important roles in promoting plant growth and inhibiting the occurrence of pathogenic bacteria [27,28]. Studies have shown that fungal communities are more influenced by host genotypes than bacterial communities in the rhizosphere [29]. The main objectives of this study were as follows: (1) to compare the composition of the fungal communities associated with the rhizosphere and endosphere of five germplasm resources of faba bean, (2) to compare the community diversity of fungi in healthy and diseased faba bean plants, and (3) to clarify the differences in fungal communities and functions between rhizosphere and endosphere in healthy and diseased faba bean plants. These results may provide theoretical guidance for future research on the prevention or control of faba bean root rot disease.

## 2. Materials and Methods

### 2.1. Study Site

Sampling was performed at a greenhouse at the Qinghai Academy of Agriculture and Forestry Sciences in Xining City (36°13′~37°28′ N, 100°52′~101°54′ E), Qinghai Province, China. A map of the study site is shown in Appendix A. Xining City has a continental plateau semi-arid climate, with mean annual temperature of 7.6 °C and mean annual precipitation of 380 mm. The terrain of Xining City is low in the northeast and high in the southwest, with an average elevation of 3137 m.

### 2.2. Sample Collection

In September 2021, when diseased faba bean plants exhibited severe root rot, healthy and diseased faba bean plants from five germplasm resources (32, 178, 211, 376, 393) were randomly selected to sample the rhizosphere soil and roots (for endosphere), which were marked as GJ and GJN in healthy faba bean and GB and GBN in diseased faba bean. Three biological replicates of each sample were collected, and there was a total of 60 samples. The images of diseased and healthy roots are shown in Appendix A. For rhizosphere soil sampling, roots were carefully dug out, and the tightly adhering soil particles on the roots after gentle shaking were brushed off and collected as rhizosphere soil. The brushed roots and rhizosphere soil were transported immediately to the laboratory in sterile plastic bags on ice and stored at 4 °C for subsequent analysis.

### 2.3. DNA Extraction, Amplification, and Illumina MiSeq Sequencing

For DNA extraction from root endosphere samples, the roots were washed successively with running water and distilled water to remove the surface-adherent soil, followed by immersion in ethanol (70%, *v*/*v*) for 4 min, 2% (*w*/*v*) NaClO_3_ solution for 1 min, and 70% ethanol for 1 min, then flushing 3 times with sterile distilled water [30]. The total DNA of the rhizosphere soil and root microbes was extracted with E.Z.N.A.^TM^ Omega Mag-Bind Soil DNA Kit (Omega Bio-Tek, Norcross, GA, USA) and E.Z.N.A.^®^ Plant DNA Kit (Omega Bio-Tek, Norcross, GA, USA) according to the manufacturer’s guidance. The concentration and quality of DNA samples were evaluated with a NanoDrop 2000 spectrophotometer (Thermo Fisher Scientific, Waltham, MA, USA) and 1% (*w*/*v*) agarose gel.

The fungal ITS region was amplified using primers ITS1F (5′-CTTGGTCATTTAGAGGAAGTAA-3′) and ITS2R (5′-GCTGCGTTCTTCATCGATGC-3′) [31]. The PCR amplicon conditions were as follows: initial denaturation at 95 °C for 3 min, denaturation at 95 °C for 30 s, annealing at 55 °C for 30 s, amplification at 72 °C for 45 s, final amplification at 72 °C for 10 min, and storage at 4 °C. Finally, the ITS libraries were sequenced on the Illumina MiSeq platform by Majorbio (Shanghai, China).

### 2.4. Bioinformatics and Statistical Analysis

The data were processed on the QIIME 1.9.1 platform [32]. The quality control, adapter trimming, and quality filtering of all sequences were performed using Fastp 0.19.6 [33]. Clean sequences were clustered into operational taxonomic units (OTUs) with a 97% identity threshold using Uparse 11, and the RDP classifier was used for taxonomic assignment against the fungal ITS in the UNITE 8.0 database [18]. The fungal sequence numbers of each sample were normalized before data analysis, and each sample obtained 30,632 effective sequences. Alpha diversity analysis was performed using Mothur 1.30.2.

A Venn diagram was constructed to assess the number of common and unique OTUs of different groups of samples. Principal coordinates analysis (PCoA) was used to analyze the similarity of fungal communities in different groups based on the Bray–Curtis distance. Analysis of similarity (ANOSIM) was performed to examine the significance of fungal communities among groups of samples. The number of permutations was set at 999. Linear discriminant effect size (LEfSe) analysis was employed to identify significant differences among the different groups of fungal communities [34]. Network analysis was applied to obtain the interaction relationship of fungal communities between rhizosphere and endosphere. The functional difference of endosphere and rhizosphere samples was predicted using FUNGuild 1.0 [35]. The box plots of the Shannon and Chao indices were drawn using Origin 2018. One-way analysis of variance (ANOVA) was carried out to calculate mean values, and Tukey’s HSD test was conducted to analyze variations among the means of alpha diversity indices at a significance level of *p* < 0.05. All data were processed with IBM SPASS Statistics, version 25.0 (IBM Corp., Armonk, NY, USA). 

## 3. Results

### 3.1. Analysis of Out Clusters

A total of 1430 fungal OTUs were shared among the samples and were assigned to 14 phyla, 44 classes, 102 orders, 218 families, 437 genera, and 694 species. Among them, 165 OTUs, or 11.54% of the total OYUs, were common for different groups (Figure 1). There were 237, 224, 258, and 146 unique OTUs, accounting for 16.57, 15.66, 18.04, and 10.21% of the total OTUs in GJ, GJN, GB, and GBN, respectively (Figure 1). The germplasm resource samples also showed shared and unique OTUs in the healthy and diseased endosphere and rhizosphere samples (Appendix A).

### 3.2. Alpha Diversity of Fungal Communities

The coverage of endosphere and rhizosphere samples was found to be between 0.998 and 1.000, which indicated that the sequencing data reflected the structure of the fungal communities of all samples. The Shannon index of healthy and diseased samples of rhizosphere soil differed significantly among germplasm resource samples (*p* < 0.05), and there was no significant difference in endosphere roots from healthy and diseased samples (*p* > 0.05) (Figure 2). The Chao index of healthy and diseased samples from rhizosphere soil and diseased endosphere root samples in faba beans from different germplasm resources differed significantly (*p* < 0.05) (Figure 2). No significant differences in the Shannon and Chao indices were found between groups (*p* > 0.05) (Appendix A).

### 3.3. Variation in Fungal Community Composition

Relative abundance higher than 0.01% at the phylum and genus levels was observed; “other” was defined as the relative abundance of less than 0.01%, and unclassified phyla and genera. The most predominant fungal phyla were Ascomycota (49.89–99.56%) and Basidiomycota (0.33–25.78%) (Figure 3 and Appendix A). Glomeromycota (0.08–1.17%) was the predominant phylum only in the GJN group samples. Olpidiomycota (0.04–1.75%) as the predominant phylum only appeared in GBN393 and GBN211 samples. Only Mortierellomycota was not the dominant phylum in the GBN group samples. The relative abundance of Mortierellomycota (20.98%) was the highest in GJN376 samples.

In all, 17, 17, 16, and 18 genera were detected in GJ, GB, GJN, and GBN group samples as the predominant genera (Figure 3). There was a significant difference in dominant genera between endosphere and rhizosphere samples. The dominant genera were similar between GJ and GB and between GJN and GBN. The dominant genera of rhizosphere samples were *Penicillium* (0.47–35.21%), *Plectosphaerella* (0.02–13.41%), *Aspergillus* (0.24–28.22%), *Chordomyces* (0.03–27.17%), *Mortierella* (0–4.48%), *Cladosporium* (0.36–3.64%), *Vishniacozyma* (0.02–7.83%), and *Tausonia* (0–1.63%) (Appendix A). Compared with GB, the unique dominant genera of GJ were *Pseudogymnoascus* (0.13–3.46%), *Trichoderma* (0–2.67%), *Paraphoma* (0–1.10%), *Trichocladium* (0–1.28%), and *Paramyrothecium* (0–1.03%) (Appendix A).

The dominant genera of endosphere samples were *Paraphoma* (3.48–91.16%), *Aspergillus* (0.06–36.05%), *Dactylonectria* (0.75–30.24%), *Apiotrichum* (0.17–21.16%), *Plectosphaerella* (0.11–18.81%), *Penicillium* (0–21.32%), *Cladosporium* (0.06–5.93%), *Clonostachys* (0–4.73%), *Vishniacozyma* (0.02–3.15%), and *Cystofilobasidium* (0–1.56%) in GJN and GBN group samples (Appendix A). The unique dominant genera in GBN were *Stachybotrys* (0–10.99%), *Rhynchogastrema* (0–5.18%), and *Olpidium* (0–1.75%) (Appendix A).

### 3.4. β-Diversity Analysis of Fungal Communities

The β-diversity of fungal communities was significantly different in the four group samples (R = 0.6174, *p* = 0.001). For different faba bean germplasm resources, GJ and GB groups showed a significant difference (R = 0.6244, *p* = 0.001; R = 0.7733, *p* = 0.001) (Appendix A), while no significant differences were seen between GJN and GBN (Appendix A), which implies that there were fewer differences in the fungal community composition in endosphere among germplasm resources samples.

In the β-diversity analysis of groups samples, there were no significant differences in the fungal communities of healthy and diseased samples in the rhizosphere groups (Figure 4A). There were significant differences between GBN and GJN, GJN and GJ, and GBN and GB (R = 0.1030, *p* = 0.026; R = 0.8767, *p* = 0.001; R = 0.9118, *p* = 0.001) (Figure 4B–D), indicating a clear distinction between the fungal community composition of rhizosphere and endosphere samples and healthy and diseased endosphere samples.

### 3.5. LEfSe Analysis of Dominant Fungal Taxa

There were 9 and 25 significantly enriched genera in the diseased endosphere and rhizosphere samples, respectively (Appendix A). The most significant genera in the GBN samples were *Paraphoma*, *Dactylonectria*, and *Epicoccum*, whereas *Penicillium* were significantly enriched in the GB samples, and these had an LDA score of >4.0. For healthy samples, there were 14 and 23 significantly enriched genera in the endosphere and rhizosphere samples, respectively (Appendix A). The most significantly enriched genera in the GJN samples were *Paraphoma*, *Dactylonectria*, *Aspergillus*, *Apiotrichum*, and *Mortierella*, while *Penicillium* and *Chordomyces* were significantly enriched in the GJ samples, and these had an LDA score of >4.0.

Comparing the endophytic fungal communities (Figure 5), there were higher proportions of *Saccharomyces*, *Lecanicillium*, and *Phialophora* in GBN than GJN, while the proportions of *Apiotrichum*, *Paramyrothecium*, *Trichothecium*, *Paraphaeosphaeria*, *Sporidiobolus*, *Xeromyces*, and *Cornuvesica* were higher in GJN. Comparing the rhizosphere fungal communities (Figure 5), there were higher proportions of *Vishniacozyma*, *Phialophora*, *Pyrenochaeta*, *Lipomyces*, *Volutella*, and *Cystofilobasidium* in GB samples, while the proportions of *Pseudogymnoascus*, *Talaromyces*, *Rodentomyces*, and *Epicoccum* were higher in GJ samples. Whether rhizosphere or endosphere samples, the significantly enriched fungal communities were different between healthy and diseased samples.

### 3.6. Network Analysis of Fungal Communities in Rhizosphere and Endosphere

There were 10 common fungal genera between healthy rhizosphere and endosphere samples: *Aspergillus*, *Paraphoma*, *Penicillium*, *Apiotrichum*, *Plectosphaerella*, *Chordomyces*, *Mortierella*, *Cladosporium*, *Vishniacozyma*, and *Verticillium* (Appendix A). In addition, 12 fungal genera were found between diseased rhizosphere and endosphere samples: *Cladosporium*, *Paraphoma*, *Penicillium*, *Clonostachys*, *Candida*, *Plectosphaerella*, *Fusarium*, *Stachybotrys*, *Aspergillus*, *Chaetomium*, *Vishniacozyma*, and *Apiotrichum* (Appendix A).

The 50 genera with the highest relative abundance were used to construct network diagrams (Figure 6). In total, 9 negative correlations and 42 positive correlations were identified from 33 fungal genera from healthy samples, which were distributed among Ascomycota, Glomeromycota, Basidiomycota, and Mortierellomycota. Ascomycota had the highest abundance in the network structure (23 nodes), accounting for 69.70%. The nodes of *Trichocladium*, *Dactylonectria*, *Metarhizium*, *Chordomyces*, and *Pseudogymnoascus* were higher in number, indicating that these taxa were more closely connected with other taxa. In addition, 6 negative correlations and 81 positive correlations were identified from 31 fungal genera from diseased samples, distributed among Ascomycota, Mortierellomycota, Basidiomycota, and Olpidiomycota (Figure 6B). The genera *Mortierella*, *Chordomyces*, *Clonostachys*, and *Clodosporium* in the diseased samples had more nodes.

### 3.7. Functional Prediction Analysis of Fungi in Different Groups

The rhizosphere and endosphere samples showed different functions (Figure 7). Undefined saprotroph was the dominant guild in the rhizosphere samples (72.47–76.99%). Undefined saprotroph and fungal parasite–undefined saprotroph were the dominant guilds in the endosphere samples, with a relative abundance of 29.05–38.08% and 20.46–47.74%, respectively. The proportion of animal pathogen–endophyte–fungal parasite–plant pathogen–wood saprotroph was higher in healthy rhizosphere samples than that in diseased rhizosphere samples. The relative abundance of saprotroph mode and saprotroph–symbiotroph mode was higher in healthy endosphere root samples than that in diseased samples. The proportion of pathotroph–saprotroph mode was higher in diseased endosphere samples than that in healthy endosphere samples.

## 4. Discussion

The unique OTU numbers of faba bean plants from different germplasm resources showed significant differences in endosphere and rhizosphere between healthy and diseased samples, indicating that different ecological groups have specific microbes. The richness and diversity of rhizosphere samples from different germplasm resources showed significant differences. In the rhizosphere of healthy and diseased samples, the fungal community diversity of germplasm 32 was greater than that of other germplasm resources. This result agrees with a previous study showing that the fungal community was influenced by the host genotype in the rhizosphere [29]. Previous studies showed that the diversity of endophytic and rhizosphere fungi was higher in diseased samples than that in healthy samples [16]; however, the results of this study are inconsistent with this. Multiple factors affect the microbial communities in soil, including plant species, climate, and soil environment [36,37]. The present study shows that the alpha diversity was not significantly different, while the β-diversity was significantly different between the rhizosphere and endosphere samples, which shows that environmental heterogeneity does not affect microbial diversity but has an impact on the structure and composition of microbial communities. In addition, the β-diversity results of four groups show that there were significant differences between healthy and diseased endosphere samples. This is consistent with the research results of *Zanthoxylum bungeanum* [38]. In addition, the β-diversity indicates that there were significant differences between the fungal communities in rhizomes of different germplasm resource samples, and there were no significant differences in the endosphere samples. This suggests that the fungal communities of different germplasm resource samples are more sensitive in the rhizosphere soil than in the roots.

Many studies have shown that endosphere microorganisms were significantly different from rhizosphere microbes, and that rhizosphere microbes were richer than endosphere microbiota [16,39], and this study confirms this conclusion. In this study, Ascomycota and Basidiomycota were the predominant phyla in all samples; the dominant phyla were similar in healthy and diseased rhizosphere samples, while healthy and diseased endosphere samples had unique dominant phyla. Glomeromycota and Olpidiomycota were the dominant phyla in the healthy and diseased endosphere samples, respectively. Arbuscular mycorrhizal fungi (AMF), which belong to the phylum Glomeromycota, have strong adaptability and tolerance to various external conditions and are beneficial for plant growth [40,41,42]. This may be the reason for the high abundance of Glomeromycota in healthy endosphere samples. Studies have shown high abundance of Olpidiomycot in soybeans with diseased roots after continuous cropping [43], which is similar to the results of this study, indicating that Olpidiomycota is a dominant phylum in diseased endosphere roots. However, the relative abundance of Mortierellomycota was higher in healthy roots, which was similar in the rhizosphere soil of healthy and diseased samples. In contrast, previous reports showed that a higher relative abundance of Mortierellomycota in healthy soil [44].

Our result showed that the *Penicillium* was more abundant in rhizosphere soil, and this was consistent with previous reports [45]. Studies indicated that some *Penicillium* spp. can produce solubilized phosphorus, siderophore, and phytohormones [46,47], which benefited plant growth. Moreover, this study found a significant enrichment of *Phialophora* in both rhizosphere and endosphere of diseased samples. The findings of various studies have demonstrated that *Phialophora* is a plant pathogen [48,49]. *Apiotrichum* is a potential antagonistic microbe against soil-borne pathogens and has a plant growth-promoting function [50,51]. This study also revealed that *Apiotrichum* was enriched in healthy samples. There was a higher relative abundance of *Paraphoma* in endosphere samples compared to rhizosphere samples, which was consistent with previous reports [52,53]. In this study, the fungal community composition showed that *Mortierella* was the dominant genus in the rhizosphere and healthy endosphere samples; however, it was not the dominant genus in the diseased endosphere samples. In addition, LEfSe analysis indicated that *Mortierella* was enriched in the healthy endosphere samples and the diseased rhizosphere samples. Previous reports showed that *Mortierella* spp. can transform phosphorus from an insoluble to a soluble form for plant uptake [54] and could significantly alleviate the diseases caused by *Fusarium oxysporum* and enhance the activities of soil sucrase and acid phosphatase [55]. *F. oxysporum* is a soil-borne disease for leguminous crops; therefore, *Mortierella* may serve as an endogenous indicator to evaluate the health of faba bean plants.

Some dominant fungal genera were found to be shared between the rhizosphere and endosphere samples; these species play important roles in plant growth and nutrient utilization. The most significantly correlated species belonged to the Ascomycota phylum. Some species were present in a higher proportion in samples, which means that these species were closely related to the entire fungal community of faba bean, such as *Trichocladium*, *Dactylonectria*, *Metarhizium*, *Chordomyces*, and *Pseudogymnoascus* in healthy samples. The positive and negative correlations indicated that some of these species may have a collaborative or a competitive relationship. Previous studies showed that the successful colonization by pathogens in plant roots is influenced by the microbial community in the soil [56]. The relationships between endosphere and rhizosphere samples might provide some clues to help us understand the link between plants and microbes.

The FUNGuild result revealed a higher relative abundance of saprotroph mode in the rhizophere samples than that in the endosphere samples. Rhizosphere saprophytic fungi can convert complex organic matter into available components that can be utilized by plants. This result indicates that the fungal communities in different habitats perform different functions. In addition, this study found a higher proportion of saprotroph–symbiotroph mode in healthy samples and a higher proportion of pathotroph–saprotroph mode in diseased samples, implying that the trophic mode of fungal communities was different between healthy and diseased samples.

## 5. Conclusions

Different faba bean germplasm resources showed shared and unique OTUs in the endosphere and rhizosphere among healthy and diseased samples. The dominant fungal phyla were Ascomycota and Basidiomycota in all samples. The most abundant genera differed between rhizosphere and endosphere samples. The diversity and richness in healthy and diseased rhizosphere of different faba bean germplasm resources differed. In addition, the richness in diseased endosphere samples of different faba bean germplasm resources differed significantly (*p* < 0.05). However, there was no significant difference in diversity between healthy and diseased endosphere roots (*p* > 0.05). The fungal community structure differed between rhizosphere and endosphere samples. Saprotrophs accounted for a larger proportion of the fungal microbes in the rhizosphere soil than in the endosphere roots. The significant correlation of key genera in diseased samples was much higher compared to that in healthy samples. This study provides a deeper understanding of the composition and diversity of fungal communities in faba bean rhizosphere and endosphere as well as a theoretical guidance for future research on the prevention or control of faba bean root rot disease.

## Figures and Tables

**Figure 1 jof-10-00084-f001:**
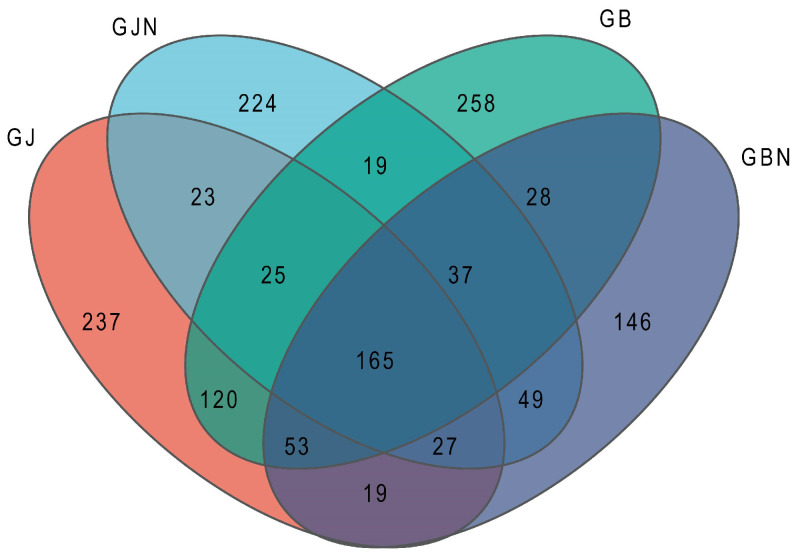
Venn diagram showing the numbers of fungal OTUs identified in GJ, GJN, GB, and GBN samples (healthy rhizosphere soil, healthy root, diseased rhizosphere soil, and diseased root, respectively).

**Figure 2 jof-10-00084-f002:**
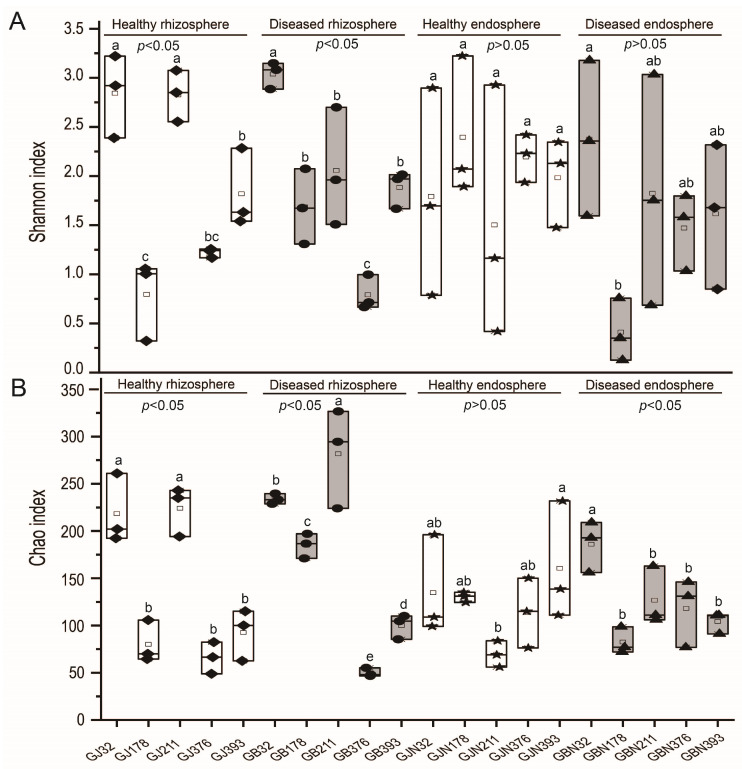
Alpha diversity indices of germplasm resource samples: (**A**) Shannon index and (**B**) Chao index. GJ, GJN, GB, and GBN represent healthy rhizosphere soil, healthy root, diseased rhizosphere soil, and diseased root samples, respectively; 32, 178, 211, 376, and 393 indicate different faba bean germplasm resources. The diamond, ellipse, pentagram, and triangle in the box plot represent samples of healthy rhizosphere soils, diseased rhizosphere soils, healthy roots, and diseased roots, respectively. Different letters represent a significant difference at the *p* < 0.05 level.

**Figure 3 jof-10-00084-f003:**
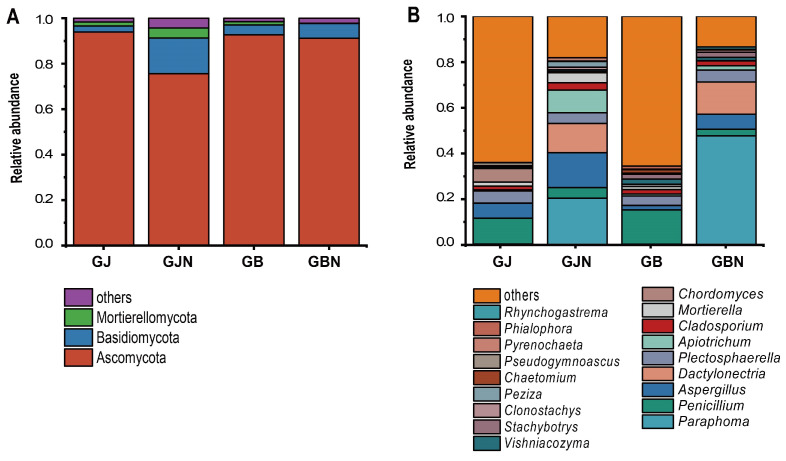
Fungal community composition of different groups at (**A**) the phylum level and (**B**) the genus level. Relative abundance of less than 0.01% was classified as “other”. GJ, GJN, GB, and GBN represent healthy rhizosphere soil, healthy root, diseased rhizosphere soil, and diseased root samples, respectively.

**Figure 4 jof-10-00084-f004:**
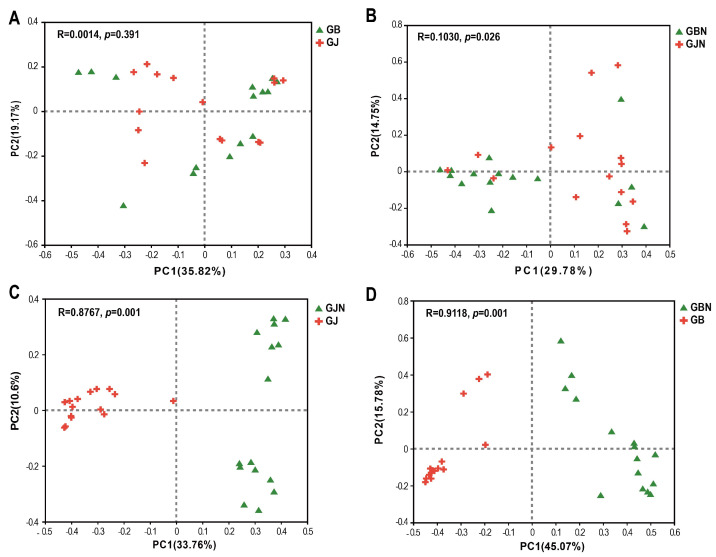
Principal coordinate analysis (PCoA) of fungal communities of rhizosphere group (**A**), endosphere group (**B**), healthy group (**C**), and diseased group (**D**) samples based on Bray−Curtis distance of OTU matrix. GJ, GJN, GB, and GBN represent healthy rhizosphere soil, healthy root, diseased rhizosphere soil, and diseased root samples, respectively.

**Figure 5 jof-10-00084-f005:**
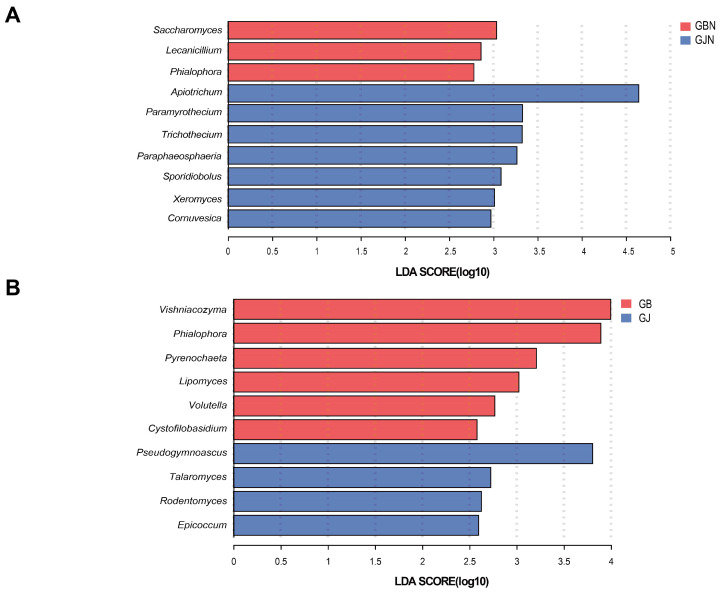
LEfSe analysis of significantly enriched fungal communities between (**A**) GBN and GJN and (**B**) GB and GJ. Fungal genera with an LDA score of >2 are shown. GJ, GJN, GB, and GBN represent healthy rhizosphere soil, healthy root, diseased rhizosphere soil, and diseased root samples, respectively.

**Figure 6 jof-10-00084-f006:**
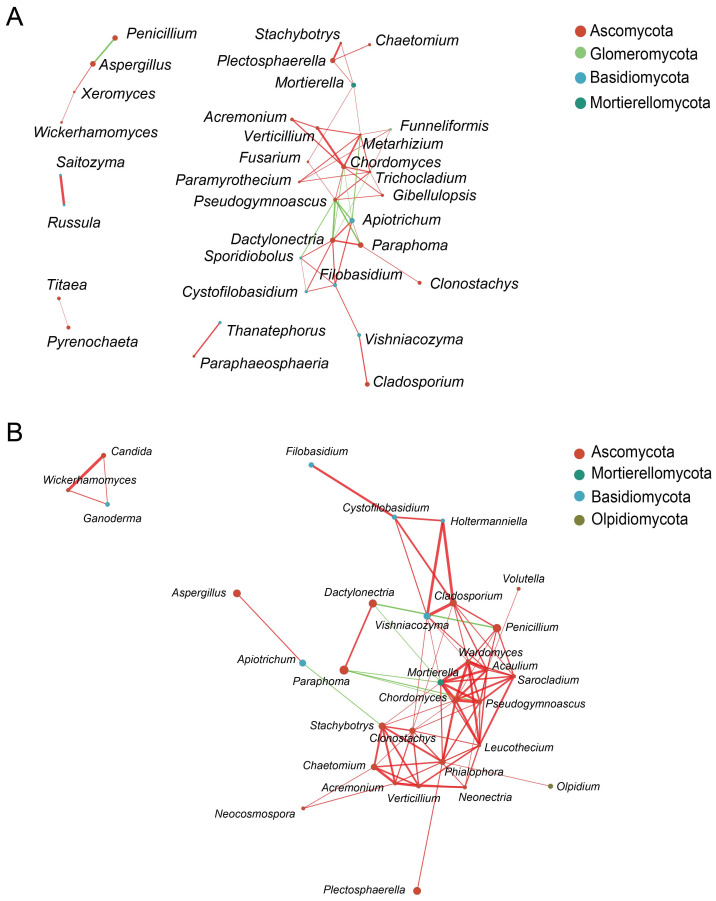
Co-occurrence network of fungi between rhizosphere and endosphere from (**A**) healthy and (**B**) diseased samples at the genera level. Only *p*-values of <0.05 and Spearman correlation coefficients of >0.5 are represented in the network. Colored dots represent phyla, with size positively correlated with abundance. Red edges indicate positive correlations, and green edges indicate negative correlations.

**Figure 7 jof-10-00084-f007:**
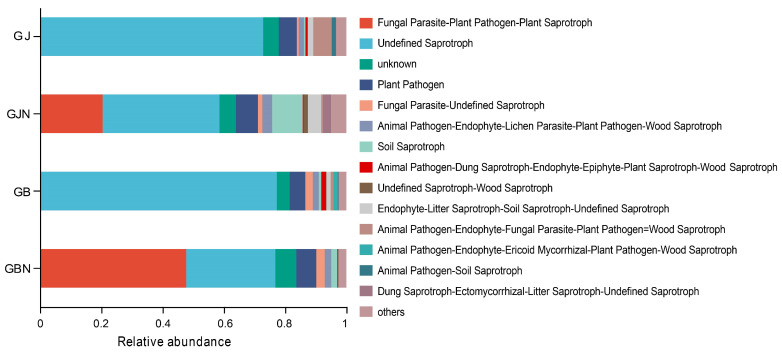
Functional analysis of fungal groups using FUNGuild. GJ, GJN, GB, and GBN represent healthy rhizosphere soil, healthy root, diseased rhizosphere soil, and diseased root samples, respectively.

## Data Availability

The original datasets presented in the study can be found online. The accession number(s) can be found here: https://www.ncbi.nlm.nih. gov/bioproject/, PRJNA1028843, accessed on 17 October 2023. All other data are provided in this article’s results section and Appendix A.

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
