# Peer review of "Comparative Analysis of Rhizosphere and Endosphere Fungal Communities in Healthy and Diseased Faba Bean Plants"

_jof, 2024, doi:10.3390/jof10010084_

Round 1
Reviewer 1 Report
Comments and Suggestions for Authors
Research expands knowledge not only theoretically but also practically. They may (as the authors themselves emphasize) constitute the basis for further equally interesting and important research.
Lines 66-70. After stating the main assumption of the research, it would be worth formulating hypotheses.
Line 79: How many samples were taken in each research case
Line 139: In the caption under Figure 1, please explain the abbreviations GJ, GJN, GB, GBN. The same in the case of Figure 3 (line 168); Figure 5 (line 236), Figure 7 (line 291)
Line 154: In the caption under Figure 2, please explain the abbreviations GJ, GJN, GB, GBN and the designations 32, 178, 211, 376, 393.
Author Response
Reviewer #1:
Comments:
Research expands knowledge not only theoretically but also practically. They may (as the authors themselves emphasize) constitute the basis for further equally interesting and important research.
Responses: We are grateful for your suggestion of our manuscript. We have modified manuscript carefully.
Lines 66-70. After stating the main assumption of the research, it would be worth formulating hypotheses.
Responses: Thank you for this suggestion. We have added the formulating hypotheses after stating the main assumption of the research. “The main objectives of this study were......These results may provides theoretical guidance for future research on the prevention or control of faba bean root rot disease.”
Line 79: How many samples were taken in each research case
Responses: Thank you. Three biological replicates of rhizosphere, endophytic, healthy and diseased samples of each germplasm resource were taken, a total of 60 samples were collected.
Line 139: In the caption under Figure 1, please explain the abbreviations GJ, GJN, GB, GBN. The same in the case of Figure 3 (line 168); Figure 5 (line 236), Figure 7 (line 291)
Responses: Thank you. We have added the abbreviations of GJ, GJN, GB, GBN under Figure 1, Figure 3, Figure 5 and Figure 7.
“Figure 1. Venn diagram showing the numbers of fungal OTUs identified of GJ, GJN, GB and GBN samples. GJ, GJN, GB and GBN represent healthy rhizosphere soil, healthy root, diseased rhizosphere soil and diseased root samples, respectively. ”
“Figure 3. Fungal community composition of different groups at the phylum level (A) and genus level (B). The relative abundance making up less than 0.01% were classified as “others”. GJ, GJN, GB and GBN represent healthy rhizosphere soil, healthy root, diseased rhizosphere soil and diseased root samples, respectively. ”
“Figure 5. The LEfSe analysis the significantly enriched fungal community between GBN and GJN (A), GB and GJ (B). Fungal genera with the LDA score of more than 2 were shown. GJ, GJN, GB and GBN represent healthy rhizosphere soil, healthy root, diseased rhizosphere soil and diseased root samples, respectively. ”
“Figure 7. Fungal functional analysis of different groups using FUNGuild. GJ, GJN, GB and GBN represent healthy rhizosphere soil, healthy root, diseased rhizosphere soil and diseased root samples, respectively. ”
Line 154: In the caption under Figure 2, please explain the abbreviations GJ, GJN, GB, GBN and the designations 32, 178, 211, 376, 393.
Responses: Thank you for this suggestion. We have added the abbreviations of GJ, GJN, GB, GBN and the designations 32, 178, 211, 376, 393 under Figure 2.
“Figure 2. Alpha diversity indices of different germplasm resources samples. (A) Shannon index; (B) Chao index. GJ, GJN, GB and GBN represent healthy rhizosphere soil, healthy root, diseased rhizosphere soil and diseased root samples, respectively. 32, 178, 211, 376 and 393 represent the different germplasm resources of faba bean.”

Reviewer 2 Report
Comments and Suggestions for Authors
This paper refers to investigation of fungal community composition and diversity of healthy and diseased samples at endosphere and rhizosphere from different germplasm resources, compared the function of samples from different samples, and explored the relationship between root and microbe of faba bean. In my opinion work is interesting in a cognitive context and contributes the new insight in knowledge of fungal communities inhabiting rhizosphere and endosphare one of the most popular cultivated plant. In general, the paper has a logical flow and it is refined in detail. The abstract well correspond with the main aspects of the work. All figures are appropriate for this type of article. However:
Figures 3 and 6 (includes also supplementary material)- I would suggest to use uniformity of colors in legend. The same genera and phyla should be marked by the same color. Why once Basidiomycota (also refers to another phyla and genera) is marked by violet, once by red as is depicted at fig. 6? Make it uniform would be more clearly.
Quite questionable is also the section 3.7 of results. The author’s approach to conclusions should be less evident and more hypothetical. Many fungi reported in diverse biotopes for endophytic or saprotrophic occurrence are better known as plant pathogens and reverse, raising questions about their actual relationships with the hosts and other plants in the biocoenosis and about the factors underlying the lifestyle shift. The ecological role of these microorganisms can change as the plant stress is a fundamental factor. I suggest to modify this section of results and rephrase the last paragraph.
Author Response
Reviewer #2:
This paper refers to investigation of fungal community composition and diversity of healthy and diseased samples at endosphere and rhizosphere from different germplasm resources, compared the function of samples from different samples, and explored the relationship between root and microbe of faba bean. In my opinion work is interesting in a cognitive context and contributes the new insight in knowledge of fungal communities inhabiting rhizosphere and endosphare one of the most popular cultivated plant. In general, the paper has a logical flow and it is refined in detail. The abstract well correspond with the main aspects of the work. All figures are appropriate for this type of article. However:
Responses: We are grateful for your suggestion of our manuscript. We have modified manuscript carefully.
Figures 3 and 6 (includes also supplementary material)- I would suggest to use uniformity of colors in legend. The same genera and phyla should be marked by the same color. Why once Basidiomycota (also refers to another phyla and genera) is marked by violet, once by red as is depicted at fig. 6? Make it uniform would be more clearly.
Responses: Thank you for your suggestion. We have uniformed the colors of the same phyla or genera in the figures 3, figure 6, figure S5 and S6 of manuscript.
Figure 3. Fungal community composition of different groups at the phylum level (A) and genus level (B). The relative abundance making up less than 0.01% were classified as “others”. GJ, GJN, GB and GBN represent healthy rhizosphere soil, healthy root, diseased rhizosphere soil and diseased root samples, respectively.
Figure 6. Co-occurrence network of fungi between rhizosphere and endosphere from healthy (A) and diseased (B) samples at genera level. Only p values less than 0.05 and Spearman correlation coefficients higher than 0.5 were constructed in the network. Different colored dots represented different phyla. The size of each node was positively correlated with the abundances of taxonomy. Red edges indicated positive correlations and green edges indicated negative correlations.
Figure S5. Fungal community composition of healthy (A) and diseased (C) samples in rhizosphere soil and of healthy (B) and diseased (D) samples in endosphere roots from different germplasms at the phylum level. The relative abundance making up less than 0.01% were classified as “others.”
Figure S6. Fungal community composition of healthy (A) and diseased (C) samples in rhizosphere soil and of healthy (B) and diseased (D) samples in endosphere roots from different germplasms at the genus level. The relative abundance making up less than 0.01% were classified as “others.”
Quite questionable is also the section 3.7 of results. The author’s approach to conclusions should be less evident and more hypothetical. Many fungi reported in diverse biotopes for endophytic or saprotrophic occurrence are better known as plant pathogens and reverse, raising questions about their actual relationships with the hosts and other plants in the biocoenosis and about the factors underlying the lifestyle shift. The ecological role of these microorganisms can change as the plant stress is a fundamental factor. I suggest to modify this section of results and rephrase the last paragraph.
Responses: Thank you for your suggestion. We have modified the section 3.7 of results and the last paragraph of discussion.
3.7. Fungal Functional Prediction Analysis of Different Groups
Undefined Saprotroph was dominant guild in the rhizosphere samples (72.47-76.99%). Undefined Saprotroph and Fungal Parasite-Undefined Saprotroph were dominant guilds in the endosphere samples with relative abundances ranging from 29.05% to 38.08% and 20.46% to 47.74%, respectively. The proportions of Animal Pathogen-Endophyte-Fungal Parasite-Plant Pathogen-Wood Saprotroph was higher in the healthy rhizosphere samples than that in diseased rhizosphere samples. The relative abundance of Saprotroph mode and Saprotroph-Symbiotroph mode were higher in healthy samples than those in diseased samples from endosphere roots. The ratio of Pathotroph-Saprotroph mode was higher in diseased endosphere samples than that observed in the healthy endosphere samples.
Discussion (the last paragraph)
The result of FUNGuild revealed that the relative abundance of saprotroph mode was higher in the rhizophere samples than that in the endosphere samples. Rhizosphere saprophytic fungi can convert complex organic matter into available components that can be utilized by plants. This result displayed that the fungal communities in different habitats performed different functions. Besides, this study showed that a higher proportion of Saprotroph-Symbiotroph mode was present in healthy samples, and the proportion of Pathotroph-Saprotroph mode was higher in the diseased samples, which implied that the trophic mode of fungal communities was different between healthy and diseased samples.

Reviewer 3 Report
Comments and Suggestions for Authors
please see attached file

Extensive editing of English language required
Author Response
Reviewer #3:
- Title
Please ensure that the indication of your institution is in accordance with the journal rules.
Responses: Thank you, we have revised the indication of our institution according to the journal rules.
1 Qinghai Academy of Agriculture and Forestry Sciences, Qinghai University, Xining 810016, China; 18293130625@163.com (J.L.); zhg-1195@163.com (G.Z); liangcheng1979@163.com (L.C.)
2 Key Laboratory of Agricultural Integrated Pest Management, Qinghai Province, Xining 810016, China
3 Key Laboratory of Qinghai-Tibetan Plateau Biotechnology(Qinghai University), Ministry of Education, Xining 810016, China
4 State Key Laboratory of Plateau Ecology and Agriculture, Qinghai University, Xining 810016, China
* Correspondence: mantou428@163.com (L.H); 13997058356@163.com(Y.L)
Abstract
- The abstract should provide an overview of the research findings. Lines 10-12 are redundant to the introduction and should be deleted. Specific results should be expressed in a numerical way rather than showing detailed methods. If you want to state that a particular genus is dominant, express it with a specific number. Numerically represent the distribution by diversity index. Readers don't want to see ambiguous results in an abstract.
Responses: Thank you very much. We have deleted the sentence of lines 10-12. Besides, we have added specific numerical values after the dominant phyla or genus and significant difference of alpha- and beta-diversity.
Abstract: In this study, the ITS approach based on Illumina MiSeq sequencing was used to assess the fungal communities of the endosphere and rhizosphere in the healthy and diseased faba bean. The findings indicated that the common dominating phyla in all samples were Ascomycota (49.89-99.56%), Basidiomycota (0.33-25.78%) and Unclassified Fungi (0.06-5.79%). Within healthy endosphere samples, Glomeromycota (0.08-1.17%) was the only predominant phylum; In diseased endosphere samples, Olpidiomycota (0.04-1.75%) was the only dominating phylum. At the genus level, the unclassified_f__Nectriaceae (24.33-86.25%) were more abundant in the rhizosphere soils while Paraphoma (3.48-91.16%) were more dominant in the endosphere roots of faba bean. Significant differences were observed in the alpha diversity of rhizosphere samples from different germplasm resources of faba bean (p < 0.05). The fungal community structures were clearly distinguished between the rhizosphere and endosphere samples, and healthy and diseased endosphere samples (P < 0.05). The unclassified_f__Nectriaceae was significantly enriched in diseased endosphere samples, whereas Apiotrichum was enriched in healthy endosphere samples. Vishniacozyma and Phialophora were enriched in diseased rhizosphere samples, while Pseudogymnoascus was enriched in healthy rhizosphere samples. The diseased samples displayed more strongly correlated genera compared with healthy samples. Saprotroph accounted for a large proportion of the fungal microbe in the rhizosphere soils than that in the endosphere roots. This study provides a better understanding of the composition and diversity of rhizosphere and endophytic fungal communities in faba bean and also provides theoretical guidance for future research on the prevention or control of faba bean root rot disease.
- unclassified_k_, unclassified_f_Fungi: This is not appropriate expression
Responses: Thank you. We have modified all the “unclassified_k_, unclassified_f_Fungi” to “Unclassified Fungi” in our manuscript.
- Manuscript needs professional English proofreading. There are too many grammatical errors.
Responses: Thank you for your suggestion. The grammatical errors of this manuscript have been proofread by colleague who has many years of experience living abroad and published multiple articles in international journals.
Materials and methods
- Study Site- Satellite or aerial maps are recommended.
Responses: Thanks for your suggestion. The satellite map of study site has been drawn as Figure S1.
Figure S1. The map showing the location of the study site
- Sample Collection- Wouldn't it be nice to attach a picture of the diseased and healthy samples?
Responses: Thank you for your suggestion. We have attached the picture of diseased and healthy roots as Figure S2.
Figure S2 The picture showed the healthy and diseased roots
- The use of greenhouse samples can also be a weakness of the study. It could be argued that the results are not representative.
Responses: Thank you. The reasons why we selected greenhouse samples was that the conditions of greenhouse were controllable, including the temperature, humidity and soil nutrition, and the microbial community structure in small-scale habitats is more similar, this excludes the influence of geographical location differences on microbial community structure, which facilitates the comparison of microbial community between different groups.
- line 80-81 The number of samples is too small.
Responses: A total of 60 samples were collected in this study. Although the number of samples is not too many, we ensure that there are three replicates for root and soil samples with healthy or diseased faba beans in each germplasm resource.
- The spread of fungi in the same space is very fast, so ecological isolation between healthy and diseased samples must be guaranteed.
Responses: Thank you. When we sampled in the greenhouse, the health samples were collected first and placed into the sealed sampling bags, then the diseased samples were collected. When we processed samples, we also processed healthy samples first and then diseased samples to prevent the spread of fungi from affecting the accuracy of the results.
Result and Discussion
- There is no need to tediously list in paragraphs the diversity information, genus and species distributions that are already presented in tables and figures. t is seriously impairing the readability of the reader.
Responses: Thank you for your suggestion. We have deleted some tedious sentences from result and discussion.
“The genera of higher relative abundance of GB group samples than GJ group samples were Stachybotrys (0-10.69%), unclassified_o__Hypocreales (0-6.19%), Chaetomium (0-7.45%), Phialophora (0-6.09%), Apiotrichum (0.05-2.48%), Clonostachys (0-2.16%), Verticillium (0-1.61%), Phoma (0-1.04%) and Neonectria (0-1.00%) (Figure S6).” was deleted.
“The genera of higher relative abundance of GJN group samples than GBN group samples were Mortierella (0-20.98%), Peziza (0-12.7%), Pyrenochaeta (0-4.70%), Filobasidium (0.30-1.70%), Titaea (0.03-1.81%), Acremonium (0-0.17%), Candida (0.02-2.02%), Chordomyces (0-2.71%), Verticillium (0-2.49%), Thanatephorus (0-2.53%), Stereum (0-1.80%), Gibellulopsis (0-1.82%), Russula (0-1.35%), Wickerhamomyces (0-1.28%), Paraphaeosphaeria (0-1.34%) and unclassified_f__Chionosphaeraceae (0-1.14%) (Figure S6).” was deleted.
”Principal co-ordinates analysis (PCoA) was used to explore the β-diversity of fungal community (Figure 4, and Figure S7). “ from 3.4. β-Diversity Analysis of the Fungal Communities was deleted.
“LEfSe was used to evaluate the significantly enriched in fungal community from different groups samples (Figure 5, and Figure S8, S9).” from 3.5. LEfSe Analysis of the Dominant Fungi Taxa was deleted.
“The Co-occurrence network was used to analyze the interaction of rhizosphere, and endosphere fungal community (Figure 6 and S10).” from 3.6. Network Analysis of Rhizosphere and Endosphere Fungal Community was deleted.
“FUNGuild was used to predicted the function of different groups (Figure 7).” from 3.7. Fungal Functional Prediction Analysis of Different Groups was deleted.
“The fungi play an important role in the root environment of faba bean. Here, we investigated fungal community composition and diversity of healthy and diseased samples at endosphere and rhizosphere from different germplasm resources, compared the function of samples from different samples, and explored the relationship between root and microbe of faba bean. “ from discussion was deleted.
- Line 354-357 It's repetitive and unnecessary. It's hard for readers to read.
Responses: Thank you for this suggestion. We have deleted these lines of sentences.
“The endophytes were closely related to rhizosphere microorganisms. The correlation network analysis provided us with a better understanding of the microbe-microbe relationship. In this present study, we analyzed the relationship between the rhizosphere and endosphere of healthy and diseased samples, respectively. ” were deleted in the discussion.
- Line 144-146. This is an unnecessary sentence in Result and discussion.
Responses: Thank you. We have deleted this sentences.
“The indices of Shannon and Chao for fungal community in endosphere and rhizosphere from healthy and diseased samples of different germplasm resources were analyzed (Figure 2). ” was deleted in the result of 3.2.
- Line 383-386. It's a logical leap to say that this is simply a diversity study. Here's why: 1) Just because a pathogen and a genus have the same name does not mean that the identified genus is pathogenic. Following the author's logic, it is like saying that E.coli O157:H7 is a pathogen and therefore all Escherichia are pathogens.
Responses: Thanks for your suggestion. The sentences expression of line 383-386 was inappropriate expression. We have modified the content in the conclusion.
inappropriate expression
Conclusion
The different germplasm resources of faba bean showed the shared and unique OTUs in the endosphere and rhizosphere among healthy and diseased samples. The dominant fungal phyla were Ascomycota, Basidiomycota and Unclassified Fungi in all samples. The most abundant genera were different in rhizosphere and endophytic samples. The diversity and richness in healthy and diseased rhizosphere of different germplasm resources faba bean differed. In addition, the richness in the diseased endosphere samples of different germplasm resources faba bean showed a significant difference (p < 0.05). However, there was no significant difference in diversity of healthy and diseased endosphere roots (p > 0.05). The fungal community structure differs between rhizosphere and endosphere samples. Saprotroph accounted for a large proportion of the fungal microbe in the rhizosphere soils than that in the endosphere roots. Significant correlation key genera in the diseased samples were much more than that in the healthy samples. This study provides a deeper understanding of the composition and diversity of rhizosphere and endophytic fungal communities in faba bean and also provides theoretical guidance for the prevention or control of faba bean root rot disease in future research.
- The author suggests that Mortierellaplays a role in the diseased samples, and Mortierella is known to be a fungus that helps plants grow in extreme environments such as coastal soils and low moisture soils. This is so well-reported that it is difficult to see how the author's claim could be convincing to the scientific community.
Responses: Thank you. The expression of “Mortierella plays a role in the diseased samples” was not appropriate. We have modified the discussion about Mortierella.
In this study, the fungal community composition result showed that the Mortierella was the dominant genus in the rhizosphere samples and healthy endosphere samples. However, Mortierella was not the dominant genus in the diseased endosphere samples. In addition, the LEfSe analysis indicated that for the healthy samples, Mortierella was enriched in the endosphere samples, but for the diseased samples, Mortierella was enriched in the rhizosphere samples. Previous reported that Mortierella spp. can transform phosphorus from an insoluble to a soluble form for plant uptake [53] and could significantly alleviate the diseases caused by Fusarium oxysporum and enhanced the activities of soil sucrase and acid phosphatase [54]. F. oxysporum was a soil-borne disease for leguminous crops, therefore, Mortierella could be served as an endogenous indicator to evaluate the health of faba bean.
- Line 320 Scientific names should appear in italics.
Responses: Thank you. We have already applied italics to Zanthoxylum bungeanum.
“This is consistent with the research results of Zanthoxylum bungeanum.”
- Line 386-388. How will this manuscript specifically benefit academia?
Responses: We think that this manuscript has two contributions for academia. Firstly, this study provides a deeper understanding of the composition and diversity of rhizosphere and endophytic fungal communities in faba bean. Secondly, this manuscript will provide theoretical guidance for the prevention or control of faba bean root rot disease in the future research.
- There is no concrete connection between results and discussion.
Responses: Thank you. We have reorganized the discussion section. The first paragraph was a discussion on alpha and beta diversity of fungal community. The second paragraph was a discussion on dominant phyla of fungal community. The third paragraph was a discussion on dominant genus of fungal community. The fourth paragraph was a discussion on correlation network diagram. The fifth paragraph was a discussion on function of fungal community.
Discussion
The unique OTUs number from different germplasm resources faba beans displayed the significant difference in endosphere and rhizosphere among healthy and diseased samples, indicating that different ecological groups have specific microbe. The richness and diversity of rhizosphere samples for different germplasm resources showed significant differences. For the rhizosphere of healthy and diseased samples, the fungal community diversity of germplasm 32 was greater than that of other germplasm resources. This result was in agreement with the previous study that the fungal community was influenced by the host genotypes in the rhizosphere [29]. Previous studies showed the diversity of endophytic and rhizosphere fungi from diseased was higher than from healthy samples [16], nevertheless, this study revealed the inconsistent result. Multiple factors affect the microbial communities in soil, for instance, plant species, climate and soil environment [36, 37]. The present study showed that the alpha diversity was not significantly different, while the β-diversity was significantly different in the rhizosphere and endosphere samples, which demonstrated that environmental heterogeneity not affected the microbial diversity, but had an impact on microbial community structure composition. In addition, the β-diversity results of four groups exhibited that there was a significant difference between healthy and diseased endosphere samples. This is consistent with the research results of Zanthoxylum bungeanum [38]. Besides, the β-diversity exhibited that the fungal community of different germplasm resources samples showed a significant difference between the rhizosphere samples, there was no significant difference in the endosphere samples, this evidence suggested that the fungal communities of different germplasm resources samples in the rhizosphere soils were more sensitive than in the roots.
Many researches proved that endosphere microorganisms were significantly different from the rhizosphere microbe, and the rhizosphere microbe were richer than the endosphere microbiota [16, 39], this study also confirms this conclusion. In this study, Ascomycota and Basidiomycota were the predominant phyla in all samples, the dominant phyla were similar in the healthy and diseased rhizosphere samples, while healthy and diseased endosphere samples had their unique dominant phyla. Glomeromycota and Olpidiomycota were the dominant phyla in the healthy and diseased endosphere samples, respectively. The arbuscular mycorrhizal fungi (AMF) belonged to the Glomeromycota phylum, which had strong adaptability and tolerance to various external environments and was beneficial taxa for plant growth [40-42]. Therefore, this may be the reason for the high abundance of Glomeromycota in healthy endosphere samples. Studies showed that Olpidiomycota exhibited a high abundance in soybeans of diseased roots after continuous cropping [43], which was similar to this study's results, indicating that Olpidiomycota was a dominant phylum in the diseased endosphere root. The relative abundance of Mortierellomycota was higher in healthy roots, nevertheless, which was similar in the rhizosphere soils of healthy and diseased samples. In contrast, previous reports showed that the relative abundance of Mortierellomycota was greater in healthy soils [44].
The species of Nectriaceae were the pathogens of crops and also were soil-borne pathogens [45, 46]. Our result showed that unclassified_f__Nectriaceae was significantly enriched in diseased endosphere samples, which implied that more unknown pathogens belonging to the Nectriaceae family harmed the health of faba bean roots and needed to be explored in future studies. Moreover, this study displayed a significant enrichment of Phialophora in both rhizosphere and endosphere of diseased samples. The findings of various studies have demonstrated that Phialophora was a plant pathogen [47, 48]. The Apiotrichum was the potential antagonistic microbe against the soil-borne pathogens and had the function of promoting plant growth [49, 50]. This study also revealed that Apiotrichum was enriched in healthy samples. The relative abundance of Paraphoma in endosphere samples exhibited a greater proportion compared to rhizosphere samples, which was consistent with previous reports [51, 52]. In this study, the fungal community composition result showed that the Mortierella was the dominant genus in the rhizosphere samples and healthy endosphere samples. However, Mortierella was not the dominant genus in the diseased endosphere samples. In addition, the LEfSe analysis indicated that for the healthy samples, Mortierella was enriched in the endosphere samples, but for the diseased samples, Mortierella was enriched in the rhizosphere samples. Previous reported that Mortierella spp. can transform phosphorus from an insoluble to a soluble form for plant uptake [53] and could significantly alleviate the diseases caused by Fusarium oxysporum and enhanced the activities of soil sucrase and acid phosphatase [54]. F. oxysporum was a soil-borne disease for leguminous crops, therefore, Mortierella could be served as an endogenous indicator to evaluate the health of faba bean.
Some shared dominant fungal genera were found in the rhizosphere and endosphere samples, these species played important roles in plant growth and nutrient utilization. The most significantly correlated species belonged to the Ascomycota phylum. Some species had more degrees in samples, which means that these species were closely related to the entire fungal community of faba bean, such as Dactylonectria, Metarhizium, Chordomyces, Pseudogymnoascus and unclassified_c__Sordariomycetes in healthy samples. The positive and negative correlations demonstrated that some of these species displayed a possible collaborative or competing relationship. Previous studies showed that the successful colonization of pathogens in the plant roots was influenced by the microbial community in the soil [55]. These relationships between endosphere and rhizosphere samples might provide some clues for the understanding of the link of plant and microbe.
The result of FUNGuild revealed that the relative abundance of saprotroph mode was higher in the rhizophere samples than that in the endosphere samples. Rhizosphere saprophytic fungi can convert complex organic matter into available components that can be utilized by plants. This result displayed that the fungal communities in different habitats performed different functions. Besides, this study showed that a higher proportion of Saprotroph-Symbiotroph mode was present in healthy samples, and the proportion of Pathotroph-Saprotroph mode was higher in the diseased samples, which implied that the trophic mode of fungal communities was different between healthy and diseased samples.

Round 2
Reviewer 3 Report
Comments and Suggestions for Authors
I believe that it has been properly revised and is ready for publication. Please consider the review results of the editor and other reviewers.
Comments on the Quality of English Languagefinal (minor) editing is required.
Author Response
Reviewer #3:
Comments and Suggestions for Authors:I believe that it has been properly revised and is ready for publication. Please consider the review results of the editor and other reviewers.
Responses: Than you for your comments and suggestion. We are also grateful for your efforts to our manuscript.
Comments on the Quality of English Language: final (minor) editing is required.
Responses: Thank you. Our manuscript have been editing by the English Language Editing Services of MDPI.
